# Caffeic Acid Phenethyl Ester Induces Vascular Endothelial Growth Factor Production and Inhibits CXCL10 Production in Human Dental Pulp Cells

Hitomi Kuramoto, Tadashi Nakanishi *, Daisuke Takegawa, Katsuhiro Mieda and Keiichi Hosaka

Department of Regenerative Dental Medicine, Institute of Biomedical Sciences, Tokushima University Graduate School, Tokushima 770-8504, Japan

* Correspondence: tnakanishi@tokushima-u.ac.jp; Tel.: +81-88-633-7340

**Abstract:** The survival rate of root non-vital teeth is lower than that of vital teeth. Therefore, to preserve the dental pulp is very important. The vascular endothelial growth factor (VEGF) is the most potent angiogenic factor involved in the vitality of dental pulp including reparative dentin formation. Caffeic acid phenethyl ester (CAPE) is a physiologically active substance of propolis and has some bioactivities such as anti-inflammatory effects. However, there are no reports on the effects of CAPE on dental pulp inflammation. In this study, we investigated the effects of CAPE on VEGF and inflammatory cytokine production in human dental pulp cells (HDPCs) to apply CAPE to an ideal dental pulp protective agent. We found that CAPE induced VEGF production from HDPCs. Moreover, CAPE induced the phosphorylation of p38 mitogen-activated protein kinase (MAPK), extracellular signal-regulated kinases (ERK), and stress-activated protein kinase/c-Jun N-terminal kinase (SAP/JNK) in HDPCs. Furthermore, CAPE inhibited C-X-C motif chemokine ligand 10 (CXCL10) production in Pam3CSK4- and tumor necrosis factor-alpha (TNF-$\alpha$)-stimulated HDPCs. In conclusion, these results suggest that CAPE might be useful as a novel biological material for vital pulp therapy by exerting the effects of VEGF production and anti-inflammatory activities.

**Keywords:** caffeic acid phenethyl ester; vascular endothelial growth factor; pulpitis; dental pulp cells; vital pulp therapy; anti-inflammatory effect

## 1. Introduction

Dental caries first occurs on the surface of the teeth and enamel and then progresses to the deep part of teeth, dentin. When caries-related bacteria reach the dental pulp, marked infiltrates of inflammatory cells are observed. In this stage, pulpitis is changed from reversible to irreversible, and dental pulp removal therapy is required. However, non-vital teeth without dental pulp are inferior to vital teeth due to a lack of sensory innervation and immune defenses, and the survival rate of root non-vital teeth is lower than of vital teeth [1]. Therefore, to preserve the dental pulp is very important. Vital pulp therapy (VPT) is a treatment to render the teeth asymptomatic while keeping the teeth vitality and functionality [2]. An ideal pulp-capping material should prevent bacterial infiltration, reduce inflammation, and form a dentin bridge [3]. The materials clinically used for dental pulp capping include calcium hydroxide (Ca(OH)$_2$) and mineral trioxide aggregate (MTA). In particular, MTA is an alkaline material that has good sealing abilities, biocompatibilities and antibacterial properties [4,5]. High success rates of VPT applying MTA have been reported by many studies [6]. However, MTA causes a mild inflammatory response [7], and thus further development of an ideal material is desired.

Dental pulp tissue is a strongly vascularized soft connective tissue [8]. Angiogenesis is a key step in the dental pulp healing which involves the reparative dentin formation [9]. Injured dental pulp cells secrete angiogenic growth factors to stimulate angiogenesis which precedes the reparative dentin formation [9]. The vascular endothelial growth factor

(VEGF), the most potent angiogenic factor [10], is known to be constitutively expressed in the dental pulp tissue [11]. A recent study showed that VEGF enhanced dental pulp cell proliferation and increased reparative dentin formation in human dental pulp [10].

In general, the initial detection of microbial pathogens is mediated by pattern recognition receptors (PRRs) against pathogen-associated molecular patterns (PAMPs). Not only immune cells but also dental pulp cells are involved in innate immunity [12]. We have already demonstrated that PRRs such as Toll-like receptors 2 (TLR2) and TLR4 are expressed in human dental pulp cells (HDPCs), and TLR2 is more clearly expressed than TLR4. [12]. Moreover, we have shown that HDPCs stimulated with Pam3CSK4, known as the TLR2 ligand, could produce C-X-C motif chemokine ligand 10 (CXCL10) which is a member of the CXC chemokine family and play an important role in the progression of pulpitis [13].

Propolis has been used in traditional medicine because of its wide spectrum of therapeutic benefits and its good safety property [14]. Caffeic acid phenethyl ester (CAPE, Figure 1) is one of the most promising physiologically active substance in honeybee propolis and has a phenolic and two-cyclic structure. CAPE is widely used as a health food and can be used safely. CAPE has many physiological effects such as anti-tumor, anti-inflammatory and antifungal [15–17]. CAPE was reported to have an inhibitory effect on tumor necrosis factor (TNF)-induced CXCL10 expression in mouse intestinal epithelial cells [18]. We have already shown that CAPE can induce VEGF, and moreover, VEGF can enhance mineralization activity in the rat odontoblastic cell line (KN-3 cells) established by limiting dilution cloning from dental papilla cells of lower incisors in Wistar rats [19]. However, the effect of CAPE on HDPCs has not yet been examined.

**Figure 1.** Structural formula of CAPE.

This study focused on the effect of CAPE on inflammation control and VEGF induction in HDPCs. The aim of this study was to investigate the effects of CAPE on inflammatory cytokines and VEGF production in HDPCs to apply CAPE to a more ideal dental pulp protective agent.

## 2. Materials and Methods

### 2.1. Cell Culture

Clinically healthy dental pulp tissue samples were obtained from non-carious teeth extracted for orthodontic reasons under informed consent at Tokushima University Hospital, Tokushima, Japan. The investigation was performed with the approval and compliance of the Ethics Committee of Tokushima University Hospital (No. 329). HDPCs were cultured in Dulbecco's Modified Eagle's Medium (Gibco, Grand Island, MI, USA) containing 10% fetal bovine serum (Sigma-Aldrich, St. Louis, MO, USA), 100 U/mL of penicillin and 100 μg/mL streptomycin (Gibco) at 37 °C in a humidified atmosphere of 5% $CO_2$. Confluent monolayers were stimulated at passages 5 to 9.

### 2.2. Reagents

CAPE was purchased from Tocris Bioscience (Bristol, UK). Pam3CSK4 (TLR2 ligand) was purchased from InvivoGen (San Diego, CA, USA). Recombinant TNF-α was obtained from Peprotech (Rocky Hill, NJ, USA).

### 2.3. Cell Proliferation Assay

HDPCs were treated with different concentrations of CAPE for 24 h and evaluated for cell proliferation activity using Cell Counting Kit-8 (DOJINDO, Tokyo, Japan). Treatment with 0.1% Triton X-100 (Wako, Osaka, Japan) was used as a positive control. Results are expressed as fold-change values relative to unstimulated control samples.

### 2.4. Enzyme-Linked Immunosorbent Assay (ELISA)

The concentrations of VEGF and CXCL10 in the cell culture supernatants were determined using enzyme-linked immunosorbent assay (ELISA) kits (Duo Set ELISA Development System; R&D Systems, Minneapolis, MN, USA) in accordance with the manufacturer's instructions.

### 2.5. Western Blot Analysis

To determine the effect of CAPE on MAPKs' phosphorylation of signal transduction molecules, Western blot analysis was performed. HDPCs were treated with CAPE for 15, 30 or 60 min and collected in RIPA lysis buffer (Santa Cruz Biotechnology, Dallas, TX, USA). The protein concentrations in lysates were quantified with a bicinchoninic acid protein assay kit (Sigma-Aldrich). An equal amount of protein was then loaded onto a 4–15% sodium dodecyl sulfate-polyacrylamide gel electrophoresis (SDS-PAGE) gel (Bio-Rad Laboratories, Hercules, CA, USA), followed by electrotransfer to a polyvinylidene difluoride membrane. The membrane was incubated with phospho-p38 MAPK antibody (Cell Signaling Technology, Danvers, MA, USA), phospho-extracellular signal-regulated kinase (ERK) antibody (Cell Signaling Technology), phospho-stress-activated protein kinase (SAP)/c-Jun N-terminal kinase (JNK) antibody (Cell Signaling Technology), p38 MAPK antibody (Cell Signaling Technology), ERK antibody (Cell Signaling Technology) and SAP/JNK antibody (Cell Signaling Technology).

The protein bands were visualized by incubation with horseradish-peroxidase-conjugated secondary antibody (Sigma-Aldrich), followed by detection using the use of ECL Prime Western Blotting Detection System (GE Healthcare, Buckinghamshire, UK). Actin levels were also assessed using an anti-actin antibody (Sigma-Aldrich) as control. The band density of blots was measured using ImageJ software (version 1.53t, US National Institutes of Health, Bethesda, MD, USA).

### 2.6. Statistical Analysis

All statistical analysis was determined by using the unpaired Student's *t* test. Differences were considered significant when the probability value was less than 5% ($p < 0.05$).

## 3. Results

### 3.1. The Effects of CAPE on Cell Proliferation in HDPCs

At first, we assessed the cytotoxicity of CAPE to HDPCs through a cell proliferation assay. The viability of HDPFs was not inhibited in the presence of CAPE (1.25–10 μg/mL) after 24 h culture (Figure 2). In our preliminary experiments to confirm the most effective concentrations for induction of VEGF and inhibition of CXCL10 production by CAPE, we found that 10 μg/mL concentrations had better effects than lower concentrations such as 5 μg/mL and 2.5 μg/mL (data not shown). Based on the above, we determined that the concentration of 10 μg/mL was optimal for this study and used it in subsequent experiments.

### 3.2. The Effects of CAPE on VEGF Production from HDPCs

We examined whether CAPE could induce VEGF production from HDPCs using ELISA. As shown in Figure 3, CAPE was able to induce VEGF production. We also investigated whether Pam3CSK4 or TNF-α could increase VEGF induction in CAPE-treated HDPCs. None of them had any effects on VEGF up-regulation.

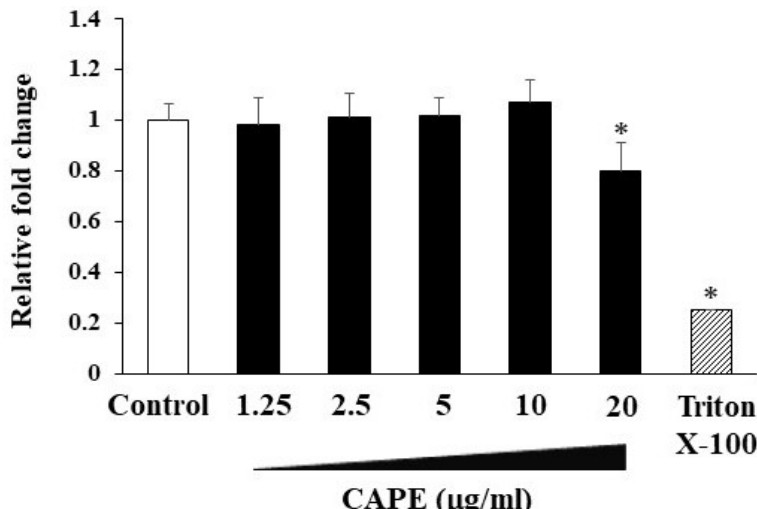

**Figure 2.** The effects of CAPE on cell viability of HDPCs. HDPCs were seeded with CAPE (1.25–20 µg/mL) for 24 h. Cell viability was assessed using Cell Counting Kit-8. Treatment with Triton X-100 was used as a positive control. Results are expressed as fold-change values relative to unstimulated control samples. Values represent the means ± SDs from representative of three independent experiments and each experiment was performed in triplicate. Asterisks indicate significant differences versus control (without CAPE) (* $p < 0.05$).

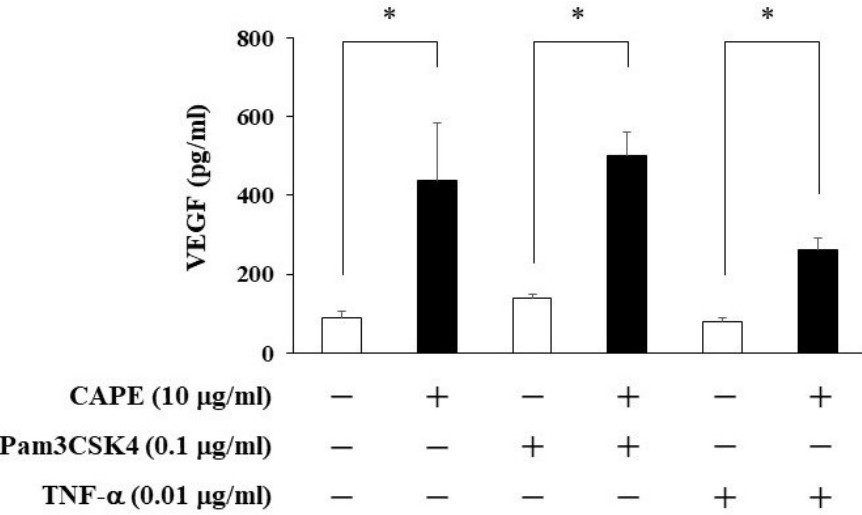

**Figure 3.** The effects of CAPE on VEGF production from HDPCs. HDPCs were treated with Pam3CSK4 (0.1 µg/mL) or TNF-α (0.01 µg/mL) and/or CAPE (10 µg/mL) for 24 h. After stimulation, the culture supernatants were collected, and the levels of VEGF were measured using ELISA kits. Values represent the means ± SDs of three independent experiments. Asterisks indicate significant differences versus control (without CAPE) (* $p < 0.05$).

*3.3. The Effects of CAPE on MAPKs' Phosphorylation in HDPCs*

To confirm the effect of CAPE in HDPCs involved in MAPKs' phosphorylation, the phosphorylation of MAPKs was analyzed using Western blot analysis. We showed that CAPE induced the phosphorylation of p38 MAPK, ERK and SAP/JNK in HDPCs at a relatively early stage (Figure 4). The graphs show the ratios of phospho-p38 MAPK, phospho-ERK, and phospho-SAP/JNK expression to total p38 MAPK, total ERK, and total SAP/JNK, respectively.

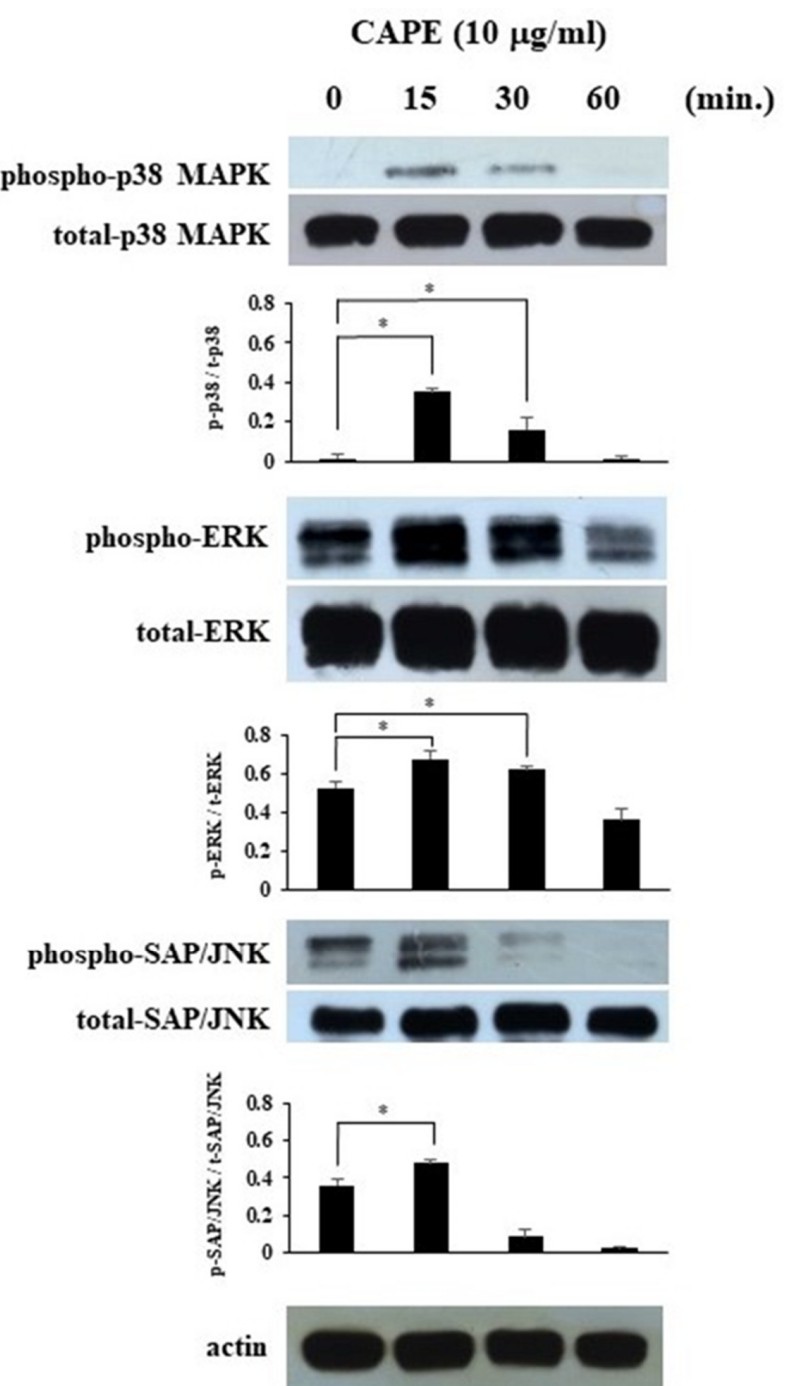

**Figure 4.** The effects of CAPE on MAPKs signaling pathways in HDPCs. HDPCs were treated with CAPE (10 µg/mL) for 15, 30, or 60 min. Western blot analysis was conducted to assess the expression of phospho-p38 MAPK, p38 MAPK, phospho-ERK, ERK, phospho-SAP/JNK, SAP/JNK and actin. Each photograph is representative of data of three separate experiments. The graphs show the ratios of phospho-p38 MAPK, phospho-ERK, and phospho-SAP/JNK expression to total p38 MAPK, total ERK, and total SAP/JNK, respectively. Values represent the means ± SDs of three independent experiments. Asterisks indicate significant differences versus without CAPE (∗ $p < 0.05$).

### 3.4. The Effects of CAPE on CXCL10 Production from HDPCs

We investigated the effects of CAPE on CXCL10 production from HDPCs. CAPE inhibited CXCL10 production in Pam3CSK4- or TNF-$\alpha$-stimulated HDPCs (Figure 5).

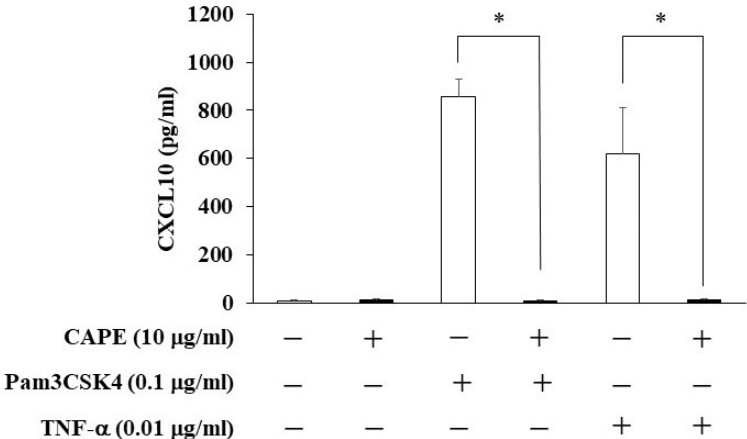

**Figure 5.** The effects of CAPE on CXCL10 production in Pam3CSK4- and TNF-$\alpha$-stimulated HDPCs. HDPCs were treated with Pam3CSK4 (0.1 µg/mL) or TNF-$\alpha$ (0.01 µg/mL) and/or CAPE (10 µg/mL) for 24 h. After treatment, the culture supernatants were collected and the levels of CXCL10 were measured using ELISA kits. Values represent the means ± SDs of three independent experiments. Asterisks indicate significant differences versus control (without CAPE) ($*$ $p < 0.05$).

## 4. Discussion

A previous study reported that VEGF was found in normal healthy dental pulps with no signs of inflammation, suggesting that VEGF was being locally produced in dental pulp tissue [11]. Moreover, in irreversible pulpitis, the expression of VEGF was strongly positive in the cells constituting the inflammatory infiltrate, whereas it was slightly, but significantly, decreased in the stromal cells [11]. VEGF is shown to enhance ALP activity and promote reparative dentin formation in HDPCs [10]. It has been reported that TNF-$\alpha$ do not affect VEGF production in dental pulp cells [20]. In addition, another study found that lipopolysaccharide (LPS) or TNF-$\alpha$ alone does not increase VEGF secretion, but LPS and TNF-$\alpha$ together had a synergistic effect on VEGF expression in HDPCs [21]. The findings of their previous reports except simultaneous stimulation are similar to the present study. Recently, we first demonstrated upregulated production of VEGF by CAPE in odontoblastic KN-3 cells [19]. In this study, VEGF production by CAPE was induced regardless of the presence of Pam3CSK4 or TNF-$\alpha$ on HDPCs. This result suggests that CAPE can possibly be applicable to dental pulp in inflammatory milieu.

MAPKs are a group of serine/threonine protein kinases that are universally expressed in mammalian cells, and have been implicated in many physiologic processes, including cell proliferation, differentiation, and death [22,23]. In this study, we confirmed that CAPE induced the phosphorylation of p38 MAPK, ERK and SAP/JNK in HDPCs. The effects of CAPE on MAPKs phosphorylation were investigated by several previous studies with various cultured cells. It was shown that CAPE induced the phosphorylation of p38 MAPK and ERK but not JNK in a rat C6 glioma cell [24]. CAPE also increased the phosphorylation of ERK, JNK and p38 MAPK to modulate the growth differentiation factor 15 (an anti-tumor gene of bladder cancer) expressions in bladder carcinoma cells [25]. In this way, the effects of CAPE on MAPKs' activation may be dependent on cell types. In a previous study, MAPKs were differentially activated by dietary chemopreventive compounds such as green tea polyphenols and involved in the transcriptional activation of antioxidant response element (ARE)-mediated reporter gene [26]. Further study will be required to investigate the detailed mechanism for the effects of CAPE on HDPCs.

When dentin is destroyed by dental caries, dental pulp cells are attacked by caries-related bacteria and bacterial products, and thus they produce pro-inflammatory mediators

such as cytokines leading to pulpitis [27,28]. Previous reports have shown that various inflammatory cytokines are expressed in dental pulp and play an important role in the development and enhancement of dental pulp inflammation [29]. CXCL10, a chemokine for recruiting activated helper T cells, is induced by pro-inflammatory stimuli and produced by dental pulp cells, which can participate in the dental pulp immune responses. In this study, we focused on the anti-inflammatory effect of CAPE, which is a bioactive substance of propolis, and investigated whether CAPE could be applied to the treatment of pulpitis. Consequently, CAPE suppressed the production of CXCL10 in HDPCs exposed to Pam3CSK4 or TNF-$\alpha$. There have been several papers mentioning the anti-inflammatory effects of CAPE. By way of example, it was shown that CAPE inhibited both TNF- and LPS-induced CXCL10 production in mouse intestinal epithelial cells [18]. In another study, CAPE inhibited the expression of interleukin (IL)-6, monocyte chemotactic protein (MCP)-1, and intercellular adhesion molecule-1 (ICAM-1) induced by IL-1$\beta$ in corneal fibroblasts [30]. Judging from these studies, CAPE is likely to have the ability of suppressing inflammation on different types of cells. We found that CAPE had no cytotoxicity for HDPCs up to 10 $\mu$g/mL by a cell proliferation assay. These findings suggest that CAPE has an anti-inflammatory effect on HDPCs in the environment of inflammation without cytotoxic impact and may be useful for the treatment of pulpitis.

For the success of vital pulp therapy, to eliminate infected bacteria and to control dental pulp inflammation is very important. It was recently reported that CAPE had antimicrobial activity against cariogenic bacteria such as *Streptococcus mutans*, *Streptococcus sobrinus*, *Actinomyces viscosus*, and *Lactobacillus acidophilus* [31]. Moreover, CAPE inhibited the formation of *S. mutans* biofilms and their metabolic activity in mature biofilms [31]. Therefore, the application of CAPE to treatment for pulpitis might be expected to eliminate a bacterial irritant directly. Our findings suggest that CAPE has the potential to be an effective material for vital pulp therapy leading to tissue regeneration via the enhancement of VEGF production. Further study using the disease model of an animal will be necessary to determinate that CAPE is applicable to treatment for dental pulp inflammation.

### 5. Conclusions

In this study, we elucidated that CAPE had a capacity to induce VEGF production in HDPCs. Moreover, CAPE inhibited CXCL10 production in Pam3CSK4- and TNF-$\alpha$-stimulated HDPCs. These findings suggest that CAPE might be useful as a novel biological material for the vital pulp therapy.

**Author Contributions:** Conceptualization, H.K. and T.N.; methodology, H.K., T.N. and D.T.; investigation, H.K., D.T. and K.M.; writing—original draft preparation, H.K.; writing—review and editing, T.N. and K.H.; supervision, T.N. and K.H.; funding acquisition, H.K. All authors have read and agreed to the published version of the manuscript.

**Funding:** This work was supported by JSPS KAKENHI (Grant Numbers: 20K23084 and 22K17041) and Research Cluster program of Tokushima University (Grant Number: 2202006).

**Institutional Review Board Statement:** The study was approved by the Ethics Committee of Tokushima University Hospital (No. 329).

**Informed Consent Statement:** Informed consent was obtained from all subjects involved in the study.

**Data Availability Statement:** Not applicable.

**Conflicts of Interest:** The authors declare no conflict of interest.

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
