# Peer review of "Caffeic Acid Phenethyl Ester Induces Vascular Endothelial Growth Factor Production and Inhibits CXCL10 Production in Human Dental Pulp Cells"

_cimb, doi:10.3390/cimb44110385_

Round 1

Reviewer 1 Report

Dear authors,

We have read with interest your manuscript describing the effects of CAPE on DPSC, especially for VEGF production and anti inflammatory effects.

To our point of view, the study is well designed, clearly described, with conclusions supported by experimental results. The whole study is convincing.

Therefore we recommend your manuscript to be accepted for publication.

Eventuallly, the Figure 1 could be modified and enriched to schematically describe the ways of action of CXCL10, Pam3CSK and TNF, and MAPK phosphorylation...

Author Response

Thank you for appreciating our study. I hope that the revised manuscript is suitable for publication in the Current Issues in Molecular Biology.

Reviewer 2 Report

Dear Authors,

Thanks for this opportunity to review the work by Hitomi Kuramoto et al. which reported that caffeic acid phenethyl ester (CAPE), an active substance of propolis, induces VEGF and inhibits CXCL10 production in human dental pulp cells (HDPCs). The topic is very interesting and important for the relative field. The results by the authors indicate that CAPE might be useful in vital pulp therapy. As the data were generated solely from ELISA or western blotting on cultured cells, the paper suffered from the weakness of evidence. My detailed concerns are listed as follows.

1)     The effect of CAPE on cell viability of HDPCs: What is the rational for choose 10 mg/ml? I suggest the authors add more concentrations, as dose-dependent viability would be more convincing.

2)     Please give the reason and/or references for the concentrations for Pam3CSK4 and TNF-a.

3)     The effects of CAPE on MAPKs signaling pathways: For the Figure 4, please add a quantification analysis base on WB data. The effects of CAPE on MAPKs signaling pathways in different time courses were never discussed by the authors. 

4)     Would it be possible for the authors to add VEGF and CXCL10 gene expression profiles to strength the evidence.

5)     The link between MAPK signaling pathway and proinflammatory cytokines in HDPCs was not tested at all. 

6)   In vivo animal test of CAPE on pulpitis model would be another point to strengthen your paper.

7)     Overall, the paper looks like a lab report, lacking logic of science. It is not a good story for the readership, in the present form.

Author Response

We are deeply grateful for the peer review process. Based on your comments, we have revised the paper and resubmitted it. The responses to each comment are as follows.

  • The effect of CAPE on cell viability of HDPCs: What is the rational for choose 10 mg/ml? I suggest the authors add more concentrations, as dose-dependent viability would be more convincing.

[Response]

We appreciate your helpful suggestion. We had already experimented with dose-dependent survival viability, so we replaced a new figure as Figure 2 (Page 4). These data demonstrated that the cytotoxic effect of CAPE (1.25-10 mg/ml) on HDPCs viability was not observed compared to control. Therefore, we used 10 mg/ml, the maximum concentration at which no cytotoxicity was observed in our experiment. In the revised manuscript, we have also revised the relevant parts in the “Materials and Methods” (Page 3, Lines 103-104) and “Results’’ section (Page 3, Lines 141-144).

  • Please give the reason and/or references for the concentrations for Pam3CSK4 and TNF-a.

[Response]

We determined the concentration based on the results of our previous experiments. That is, we referred to reference 12 (Hirao et al.) for Pam3CSK (0.1 mg/ml) and reference 13 (Adachi et al.) for TNF-a (0.01 mg/ml).

  • The effects of CAPE on MAPKs signaling pathways: For the Figure 4, please add a quantification analysis base on WB data. The effects of CAPE on MAPKs signaling pathways in different time courses were never discussed by the authors.

[Response]

We inserted additional graphs characterizing the results of western blot quantitative analysis in Figure 4 (Page 5) These data demonstrated the ratios of phospho-p38 MAPK, phospho-ERK, and phospho-SAP/JNK expression to total p38 MAPK, total ERK, and total SAP/JNK, respectively. We showed that CAPE induced the phosphorylation of MAPKs in HDPCs at a relatively early stage (15 minutes). In the revised manuscript, we have also amended the relevant parts in the “Materials and Methods” (Page 3, Lines 132-133) and “Results’’ section (Page 5, Lines 201-204).

  • Would it be possible for the authors to add VEGF and CXCL10 gene expression profiles to strength the evidence.

[Response]

Unfortunately, we do not have data on their gene expression profiles. In this experiment, we considered the protein changes to be important.

  • The link between MAPK signaling pathway and proinflammatory cytokines in HDPCs was not tested at all.

[Response]

We have already investigated whether Pam3CSK4 is involved in the phosphorylation of MAPKs in HDPCs. The results confirmed that Pam3CSK4 induced the phosphorylation of MAPKs. (Reference: Hirao, K.; Yumoto, H.; Nakanishi, T.; Mukai, K.; Takahashi, K.; Takegawa, D.; Matsuo, T. Tea catechins reduce inflammatory reactions via mitogen-activated protein kinase pathways in toll-like receptor 2 ligand-stimulated dental pulp cells. Life sciences 2010, 86, 654-660, doi:10.1016/j.lfs.2010.02.017.)

In addition, it has been shown that TNF-a induced the phosphorylation of MAPKs in HDPCs. (Reference: Zhao, Y.; Wang, C.L.; Li, R.M.; Hui, T.Q.; Su, Y.Y.; Yuan, Q.; Zhou, X.D.; Ye, L. Wnt5a promotes inflammatory responses via nuclear factor κB (NF-κB) and mitogen-activated protein kinase (MAPK) pathways in human dental pulp cells. The Journal of biological chemistry 2014, 289, 21028-21039, doi:10.1074/jbc.M113.546523.) In this study, we showed that CAPE induced the phosphorylation of MAPKs in HDPCs.

Based on the above, we examined the expression of CXCL10 using MAPK pathway signaling inhibitors in HDPCs treated with Pam3CSK4 (or TNF-a) and CAPE, but no changes were observed (data not shown in this paper). Therefore, at present, we do not know the detailed mechanism by which CAPE causes the inhibition of CXCL10 production in Pam3CSK4- or TNF-a-stimulated HDPCs. This will be an issue for future study.

  • In vivo animal test of CAPE on pulpitis model would be another point to strengthen your paper.

[Response]

We also agree with your suggestion. We should consider using CAPE for animal models of pulpitis at the next stage, although we were not able to perform it in this study.

  • Overall, the paper looks like a lab report, lacking logic of science. It is not a good story for the readership, in the present form.

[Response]

Thank you for pointing this out. We have done the best we can with the current situation.

Again, thank you for giving us the opportunity to strengthen our manuscript with your valuable comments and queries.

Reviewer 3 Report

The manuscript showed the Caffeic Acid Phenethyl Ester Induces Vascular Endothelial Growth Factor Production and Inhibits CXCL10 production in Human Dental Pulp Cells. The authors have demonstrated that CAPE has the ability to induce VEGF production in HDPCs. It was also shown that CAPE inhibits CXCL10 production in HDPC stimulated with Pam3CSK4 and TNF-α. However, there are a few suggestions to improve it as follows.

Comment;

1. How did you determine the concentration of CAPE used in the experiment? It must be demonstrated that the concentration of 10 ug of CAPE is the concentration with the best effect.

2. After HDPC was treated with CAPE, the concentrations of VEGF and CXCL10 were measured in cell culture supernatants. Are there any changes in mRNA levels?

3. The authors reported that CAPE has anti-inflammatory activities. Does CAPE affect the expression of other proinflammatory cytokines, including TNF-a?

4. In Figures 3 and 5, inflammation was induced by TNF-a. However, pulpitis caused by caries is induced by TNF-a-like inflammatory cytokines by caries-causing bacteria (S. Mutans) and/or their LPS. Therefore, to support the authors' claim, it should be confirmed whether CAPE has an anti-inflammatory effect in the inflammatory response induced by LPS.

5. In dental pulp, inflammatory cytokines induced by TNF-a or LPS are mediated by Nfkb. Therefore, it is necessary to confirm the expression of Nfkb in Figure 4. In addition, it is necessary to add the results of VEGF and CXCL10 protein levels over time after treating with CAPE.

6. Figure 4 shows that CAPE inhibits the p38 MAPK/JNK/Erk pathway. However, it should show changes in the p38 MAPK/JNK/Erk pathway when treated with TNF-a or Pam3CSK4 and treated with CAPE.

Author Response

We are deeply grateful for the peer review process. We have revised the paper based on the comments received and resubmitted it. The responses to each comment are as follows.

  1. How did you determine the concentration of CAPE used in the experiment? It must be demonstrated that the concentration of 10 ug of CAPE is the concentration with the best effect.

[Response]

We appreciate your helpful suggestion. We had already experimented with dose-dependent survival viability, so we replaced a new figure as Figure 2 (Page 4). These data demonstrated that the cytotoxic effect of CAPE (1.25-10 mg/ml) on HDPCs viability was not observed compared to control. Therefore, we used 10 mg/ml, the maximum concentration at which no cytotoxicity was observed in our experiment. In the revised manuscript, we have also revised the relevant parts in the “Materials and Methods” (Page 3, Lines 103-104) and “Results’’ section (Page 3, Lines 141-144).

  1. After HDPC was treated with CAPE, the concentrations of VEGF and CXCL10 were measured in cell culture supernatants. Are there any changes in mRNA levels?

[Response]

Unfortunately, we do not examine the changes in mRNA levels. In this experiment, we put more emphasis on protein changes.

  1. The authors reported that CAPE has anti-inflammatory activities. Does CAPE affect the expression of other proinflammatory cytokines, including TNF-a?

[Response]

Preliminary examination of IL-6 and CCL20 expression showed that CAPE did not induce IL-6 and CCL20 production and had no anti-inflammatory effect.

  1. In Figures 3 and 5, inflammation was induced by TNF-a. However, pulpitis caused by caries is induced by TNF-a-like inflammatory cytokines by caries-causing bacteria (S. Mutans) and/or their LPS. Therefore, to support the authors' claim, it should be confirmed whether CAPE has an anti-inflammatory effect in the inflammatory response induced by LPS.

[Response]

In our research to date, we have reported that clear expressions of TLR2 and a faint expression of TLR4 in HDPCs by reference 12 (Hirao et al.).

Therefore, we used the Pam3CSK4 as TLR2 ligand for this experiment rather than the LPS as TLR4 ligand. We have added this comment in “Introduction” section (Page 2, Lines 55-56) and hope that this description clarifies the points we attempted to make.

  1. In dental pulp, inflammatory cytokines induced by TNF-a or LPS are mediated by Nfkb. Therefore, it is necessary to confirm the expression of Nfkb in Figure 4. In addition, it is necessary to add the results of VEGF and CXCL10 protein levels over time after treating with CAPE.

[Response]

CAPE is known as an inhibitor of NF-kB, although we have not examined it in this time. (Reference: Natarajan, K.; Singh, S.; Burke, T.R., Jr.; Grunberger, D.; Aggarwal, B.B. Caffeic acid phenethyl ester is a potent and specific inhibitor of activation of nuclear transcription factor NF-kappa B. Proceedings of the National Academy of Sciences of the United States of America 1996, 93, 9090-9095, doi:10.1073/pnas.93.17.9090.)

As to the time course, we selected and performed the 24 hours treatment of CAPE, based on the results of our preliminary experiments,

  1. Figure 4 shows that CAPE inhibits the p38 MAPK/JNK/Erk pathway. However, it should show changes in the p38 MAPK/JNK/Erk pathway when treated with TNF-a or Pam3CSK4 and treated with CAPE.

[Response]

In this study, we showed that CAPE induced the phosphorylation of MAPKs in HDPCs as shown in Figure 4.

We have already investigated whether Pam3CSK4 is involved in the phosphorylation of MAPKs in HDPCs. The results confirmed that Pam3CSK4 induced the phosphorylation of MAPKs. (Reference: Hirao, K.; Yumoto, H.; Nakanishi, T.; Mukai, K.; Takahashi, K.; Takegawa, D.; Matsuo, T. Tea catechins reduce inflammatory reactions via mitogen-activated protein kinase pathways in toll-like receptor 2 ligand-stimulated dental pulp cells. Life sciences 2010, 86, 654-660, doi:10.1016/j.lfs.2010.02.017.)

In addition, it has been shown that TNF-a induced the phosphorylation of MAPKs in HDPCs. (Reference: Zhao, Y.; Wang, C.L.; Li, R.M.; Hui, T.Q.; Su, Y.Y.; Yuan, Q.; Zhou, X.D.; Ye, L. Wnt5a promotes inflammatory responses via nuclear factor κB (NF-κB) and mitogen-activated protein kinase (MAPK) pathways in human dental pulp cells. The Journal of biological chemistry 2014, 289, 21028-21039, doi:10.1074/jbc.M113.546523.)

Based on the above, we examined the expression of CXCL10 using MAPK pathway signaling inhibitors in HDPCs treated with Pam3CSK4 (or TNF-a) and CAPE, but no changes were observed (data not shown in this paper). Therefore, at present, we are unable to explain the detailed mechanism by which CAPE causes the inhibition of CXCL10 production in Pam3CSK4- or TNF-a-stimulated HDPCs. In addition, since CAPE has an aspect as an NF-κB inhibitor, it may be partially involved in the suppression of CXCL10 expression. This will be an issue for future study.

Again, thank you for giving us the opportunity to strengthen our manuscript with your valuable comments and queries.

Round 2

Reviewer 2 Report

For Figure 4, Pls add mean+-SD bar charts for the western blotting.

Author Response

Response to the Reviewer 2’s Comments

Thank you for your valuable comment. Based on your comments, we have revised the paper and resubmitted it. The responses to the comment are as follows.

[Comments and Suggestions for Authors]

For Figure 4, Pls add mean+-SD bar charts for the western blotting.

[Response]

We added the means ± SD bars to the graph characterizing the results of the western blot quantitative analysis and inserted them in Figure 4 (Page 5). In the revised manuscript, we have also amended the figure 4 caption (Page 5, Lines 248-249).

Reviewer 3 Report

There is still a lack of replies to comments.

Data should be presented to determine the optimal concentration for VEGF and CXCL10 expression, not the concentration to determine the effect of CAPE on cell proliferation.

Author Response

Response to the Reviewer 3’s Comments

Thank you for your valuable comment. The responses to the comment are as follows.

[Comments and Suggestions for Authors]

There is still a lack of replies to comments.

Data should be presented to determine the optimal concentration for VEGF and CXCL10 expression, not the concentration to determine the effect of CAPE on cell proliferation.

[Response]

Based on the results of cytotoxicity studies, we tentatively decided on a CAPE concentration of 10 mg/ml, which was used in our previous studies (reference 19; Kuramoto et al.). This CAPE concentration has also been applied in other studies such as “Wang, X.; Stavchansky, S.; Zhao, B.; Bynum, J.A.; Kerwin, S.M.; Bowman, P.D. Cytoprotection of human endothelial cells from menadione cytotoxicity by caffeic acid phenethyl ester: the role of heme oxygenase-1. Eur J Pharmacol 2008, 591, 28-35, doi:10.1016/j.ejphar.2008.06.017.”, although the cell types are different.

Furthermore, as a preliminary experiment for this study, we confirmed that 10 μg/ml was the most effective concentration of CAPE for inducing VEGF and suppressing CXCL10 production.

Therefore, in this study, this concentration was selected for the experiment.

Round 3

Reviewer 3 Report

  1. I have checked your paper, and thank you for your explanation. However, since the cells are different, I believe that an experiment to confirm the effective concentration in Pulp cells is necessary. How did you determine the concentration of CAPE used in the experiment? It must be demonstrated that the concentration of 10 ug of CAPE is the concentration with the best effect, not the concentration for cell viability. 

Author Response

Response to the Reviewer 3’s Comments

Thank you for your valuable comment. Based on your comments, we have revised the paper and resubmitted it. The responses to the comment are as follows.

[Comments and Suggestions for Authors]

  1. I have checked your paper, and thank you for your explanation. However, since the cells are different, I believe that an experiment to confirm the effective concentration in Pulp cells is necessary. How did you determine the concentration of CAPE used in the experiment? It must be demonstrated that the concentration of 10 ug of CAPE is the concentration with the best effect, not the concentration for cell viability.

[Response]

As mentioned in our recent response, we confirmed the most effective concentrations for induction of VEGF and inhibition of CXCL10 production by CAPE from preliminary experiments.

Specifically, concentrations of less than 10 mg/ml, such as 5 mg/ml and 2.5 mg/ml, showed weaker VEGF induction and suppression of CXCL10 production than 10 mg/ml.

Therefore, the experimental concentration of 10 mg/ml was finally determined based on the results of cell viability assays and preliminary experiments examining the effect of concentration.

We have added a description of the concentration determination to the "Results" section (Page 3, Lines 141-146).

Round 4

Reviewer 3 Report

Thanks for your reply.